# Rational design of a survey protocol for avocado sunblotch viroid in commercial orchards to demonstrate pest freedom

D. B. Bonnéry[1]*, L. -S. Pretorius[2], A. E. C. Jooste[3], A. D. W. Geering[2], C. A. Gilligan[1]

1 Department of Plant Sciences, University of Cambridge, Cambridge, United Kingdom, 2 Centre for Horticultural Science, Queensland Alliance for Agriculture and Food Innovation, The University of Queensland, St Lucia, QLD, Australia, 3 Agricultural Research Council-Tropical and Subtropical Crops, Mbombela, South Africa

* dbb31@cam.ac.uk

**Data Availability Statement:** All relevant data and code are both within the manuscript Supporting Information files and on a publically available website. The website explaining how to run the

## Abstract

Avocado sunblotch viroid (ASBVd) is a subcellular pathogen of avocado that reduces yield from a tree, diminishes the appearance of the fruit by causing unsightly scarring and impedes trade because of quarantine conditions that are imposed to prevent spread of the pathogen via seed-borne inoculum. For countries where ASBVd is officially reported, permission to export fruit to another country may only be granted if an orchard can be demonstrated to be a pest free production site. The survey requirements to demonstrate pest freedom are usually defined in export protocols that have been mutually agreed upon by the trading partners. In this paper, we introduce a flexible statistical protocol for use in optimizing sampling strategies to establish pest free status from ASBVd in avocado orchards. The protocol, which is supported by an interactive app, integrates statistical considerations of multistage sampling of trees in orchards with a RT-qPCR assay allowing for detection of infection in pooled samples of leaves taken from multiple trees. While this study was motivated by a need to design a survey protocol for ASBVd, the theoretical framework and the accompanying app have broader applicability to a range of plant pathogens in which hierarchical sampling of a target population is coupled with pooling of material prior to diagnosis.

## Introduction

Worldwide, the avocado industry is growing at an unprecedented rate, faster than for any other tropical fruit. Modelling suggests that by 2030, global production will have tripled compared with 2010 levels and the avocado (*Persea americana*) will become the most traded tropical fruit, overtaking both pineapple and mango in quantity terms [1]. With this growing trade, closer scrutiny will be placed on the risks of spreading pests and pathogens in fresh fruit. Avocado sunblotch viroid (ASBVd) is a major trade concern for many countries as the pathogen may be transmitted at rates of 86–100% in seed from -asymptomatically-infected trees [2].

Free trade is governed by bilateral or multilateral agreements that typically are subject to sanitary or phytosanitary conditions to protect human, animal or plant life and health. The

code is available at:https://gilligan-epid.uniofcam. dev/asbvddetect/asbvd-detect/. An R package has been built. An R package is a zipped folder, containing subfolders. One subfolder named R contains functions, the other named demo a single code that can reproduce the results, and a third folder data contains the data in R format. This data was created from spreadsheets used by the experimnetors to collect the data. Those spreadsheets are in the folder inst/extdata and the programs to read them are in the folder R. The package has been attached to this paper and is also available online at https://gitlab.developers.cam.ac. uk/gilligan-epid/asbvddetect/asbvd-detect-r.

**Funding:** This project has been funded by Hort Innovation (https://www.horticulture.com.au/), using the avocado research and development levy and contributions from the Australian Government under Project AV18007 – Avocado sunblotch viroid survey. Hort Innovation is the grower owned, not-for-profit research and development corporation for Australian horticulture. The grant was received by A. G. The funders had no role in study design, data collection and analysis, decision to publish, or preparation of the manuscript.

**Competing interests:** The authors have declared that no competing interests exist.

principles guiding application of biosecurity measures within a free trade environment are codified in 'The World Trade Organization (WTO) Agreement on the Application of Sanitary and Phytosanitary Measures (SPS Agreement)', which aims to minimize the arbitrariness of decisions and to encourage consistent decision-making. The WTO itself does not develop the international standards but leaves this to scientists and government experts in plant protection [3]. Responsibility to develop international phytosanitary standards is delegated to the 'Commission on Phytosanitary Measures', which is the governing body of the 'International Plant Protection Convention (IPPC)'. These standards are known as 'International Standards for Phytosanitary Measures' (ISPMs), and as of January 2022, there were 44 adopted ISPMs [4].

A common thread of all the ISPMs is that any decision on phytosanitary measures must be justified by scientific evidence. It is recognised that when plants or plant products are imported into a country, the risk of introduction of pests (including pathogens) cannot be entirely removed and therefore a policy of 'managed risk' should be applied (clause 1.3, ISPM 1). There is provision for an exporting country to declare an area as being pest free or a place of low pest prevalence (clause 2.3, ISPM 1) but this claim must be supported by survey data (clause 1.1, ISPM 10). The scale of a pest-free area can vary in size, from an entire nation to an individual farm and when referring to the latter, the term 'pest-free production site' is used. Several criteria need to be satisfied for a farm to be designated a pest free production site including geographic isolation of the farm, lack of natural or artificial infection pathways by which the pathogen could be introduced onto the farm, and the availability of sufficiently sensitive methods for detection of the pest (clause 2.1, ISPM 10). Systems need to be established to establish pest freedom in the first place and then maintain this status (clause 2.2, ISPM 10) but no guidance is provided as to what these systems should be, other than they are normally set by the national plant protection organization. Here we introduce and test a statistical protocol to designate pest freedom from ABSVd in a commercial orchard. The methods are scalable to multiple orchards and adaptable for other host-pathogen systems.

Avocado sunblotch viroid has many biological characteristics that are conducive to the creation of pest free production sites, as recently reviewed by Kuhn et al. [5] and Saucedo Carabez et al. [6]. The viroid has no arthropod vector and in nature its host range is restricted to avocado. While pollen transmission is recorded at a low rate, it only results in infection of the seed embryo and not the maternal tree [7, 8]. There are few spatiotemporal analyses of ASBVd epidemics but in the only notable study, the infection status of avocado trees in the National Germplasm Repository-Miami, Florida was monitored for many years, and field spread shown to be slow [9]. Across the entire germplasm collection, the incidence of infected trees only increased by 4.7% over a 9-year period [9]. In the field containing the oldest trees, 12 of the 14 newly infected trees were adjacent to previously infected trees, suggesting transmission by natural root grafting [9].

Reverse transcription-(RT) qPCR is now the preferred diagnostic method for ASBVd in many countries around the world [5]. However, there are variations in the RT-qPCR assay format and sampling protocol used for ASBVd diagnosis between countries. In New Zealand, the diagnostic standard specifies that when testing asymptomatic trees, a total of ten leaves should be collected from the four compass points of the tree at the height of a standing person, ideally taking single leaves from separate branches [10]. In Florida, six leaves are sampled from each tree, four from around the base and two from the top of the tree [11]. The sampling protocol in South Africa specifies that 20 to 24 leaves should be collected from all the main branches of a tree when testing individual asymptomatic trees, and eight leaves per tree when pooling three trees in one sample. In these sampling protocols, the premise for collecting multiple leaves from a tree is a presumption of an uneven distribution of the viroid within the tree, particularly between different branches [12, 13].

Surveying to demonstrate pest freedom in an avocado orchard is not a trivial task. In Australia, an orchard may contain 30,000 trees, which at maturity are *c.* $10 \times 5$ m in dimensions. The expense of surveying dramatically increases if there is a requirement to collect leaves from any layer other than at ground height, with additional safety issues associated with using ladders or mechanical lifters. Batch-testing methods must be employed to test the thousands of leaves that may be collected.

The objective of this study was to develop a tool to optimize, with respect to cost, a sampling protocol to demonstrate pest freedom from ASBVd at the production site level, given a set of constraints required by regulatory authorities. These constraints are expressed in terms of detecting the viroid with a given level of confidence in an orchard with a specified prevalence, maximum size and efficacy of detection of the viroid, where the latter is referred to as 'method sensitivity' [14]. Optimization is achieved by calculating the minimal sample size needed to satisfy the regulatory constraints. The method sensitivity is conditional on the laboratory diagnostic protocol that is used and needs to be estimated afresh and agreed by regulatory authorities for different diagnostic protocols [14]. Formulae exist to obtain the optimal allocation when a pathogen is either uniformly distributed or occurs in a predictable pattern within a plant, the pattern of infection across the field is completely random and samples from different plants are tested individually [15]. For example, the optimal number of trees to be tested can be determined using a simple stratified sampling optimization for disease freedom [16]. However, if any one of these conditions are not met, then the formula for estimating the risk of no detection inevitably changes.

In this paper, we describe experiments that were done to determine the method sensitivity for detection of ASBVd in an orchard by RT-qPCR for use in the statistical protocol. The Methods section of this paper formalizes the definition of the regulatory constraints for optimal sampling and describes the experiments that were conducted to support optimal sampling to test for freedom of infection. The experimental results and the protocol for optimal allocation, in terms of sample distribution and pooling of leaf samples, are summarized in the Results section along with general recommendations for the protocol. Finally, a user-friendly software application is presented to allow selection of the optimal allocation of samples to demonstrate pest freedom.

## Materials and methods

### Field sites and PCR diagnostic protocols

Experiments were undertaken in South Africa, where there is ready access to ASBVd-infected trees and in Australia, the initial target for the sampling protocol but where the pathogen is extremely rare and known to be present in only four avocado 'Hass' trees at a single orchard in South-east Queensland (location protected for privacy reasons). A 'Hass' orchard in the eastern region of the Mpumalanga province was used in experiments done in South Africa, where an estimated 7% of the trees are infected with ASBVd, all asymptomatically.

The laboratories in Australia and South Africa are the reference testing laboratories for ASBVd in their respective countries and utilize TaqMan® and SYBR Green™ detection methods, respectively. The two methods described below are both highly sensitive and provide comparative results (Ct values) when the filter disc extraction method is used for RNA extraction and parallel tests are done using the same starting material [17].

In Australia, an 8-mm biopsy punch was used to obtain leaf discs that were placed in 2 mL Safe-Lock tubes (Eppendorf). Leaves were freeze-dried overnight and then stored at -80°C until testing. RNA was extracted using the filter disc extraction method and then qRT-PCR done as described by Pretorius et al. [17], using the TaqMan Fast Virus 1-Step Master Mix

(Thermo Fisher Scientific). The Ct value was obtained within 40 cycles, and thereafter, the cycling process was stopped, and the value reported as undetermined.

In South Africa, a similar method to that described above was used to obtain the leaf discs and double-stranded RNA (dsRNA) was extracted using the cellulose column-chromatography method of Luttig and Manicom [18]. A one step qRT-PCR assay for ASBVd was done using a qPCRBIO SyGreen 1-Step Go Lo-Rox kit (PCRBIOSYSTEMS, UK) according to the manufacturer's instructions and primers (5'– AGAGAAGGAGGAGTCGTGGTGAAC –3'; 5'– TTCCCATCTTTCCCTGAAGAGAC –3') were used, each present at a final concentration of 400 nM in a 12.5 μl reaction volume. Reverse transcription was done at 50˚C for 10 min, followed by polymerase activation at 95˚C for 2 min. The cycling conditions included reverse transcription at 50˚C for 10 min, followed by polymerase activation at 95˚C for 2 min. The PCR step included 35 cycles with a denaturation step at 95˚C for 5 sec and an annealing step at 56˚C for 30 sec, followed by a melt analysis. All reactions were run on a Rotor-Gene Q machine (QIAGEN) and the accompanying software (v. 2.3.1) used for the analyses.

## Experiments 1 and 2: Leaf, branch (and octant) and tree effects on viroid titer

Experiment 1 was designed to investigate patterns of ASBVd infection across the tree to test in principle if some parts of the leaf are preferentially infected. Leaf samples were collected on 24 August 2020 from two trees in the Australian orchard that were asymptomatically infected. As negative controls, leaves were also collected from two trees that were classified as uninfected after repeated testing by qRT-PCR prior to this experiment. Each tree was divided into eight octants to ensure even sampling of the tree, covering the top and bottom of the tree and all four sides. A total of 96 leaves, all of which were asymptomatic, were collected per infected and uninfected tree. Twelve leaves were collected per octant, taking note of the main branches from which each leaf was collected. Bags were labelled accordingly and leaves from the same octant and branch were placed in the same bag. Samples were transported to the laboratory in a cooler box containing ice-bricks. The Ct values obtained from the qRT-PCR analyses were summarized in a data frame containing the following variables: tree, branch, quadrant, leaf, biological replicate index, technical replicate index and Ct measurement [S1 Fig 1 in S1 File].

Tree status and tree and branch effects were treated as fixed effects and leaf and position of leaf disc were taken as random effects. The octant and disc effects were analyzed as both nested and crossed effects. A crossed (i.e., interaction) effect would indicate systematic differences in the viroid at different positions on the tree or within a leaf, for example, by preferential accumulation in upper octants or at the tip of the leaf.

Experiment 2 was designed to test for systematic variation in titer of ASBVd within a leaf. samples were collected from a third tree in Australia, which was almost entirely asymptomatic except for a small branch with a cluster of variegated leaves. Five each of symptomatic and asymptomatic leaves were collected and each leaf was sampled using the biopsy punch at eleven consistent locations. RNA extracts were tested by RT-qPCR in duplicate and Ct values for each leaf disc averaged [Fig 1; S1 Fig 2 in S1 File].

Potential patterning of infection within trees by which we mean the identification of areas that are consistently preferentially infected was tested by comparing Ct values from leaves on different branches and octants in a sample of healthy and infected trees. General mixed models were used with likelihood ratio tests (LRTs) [19] to estimate the variances in ASBVd titer between trees, between branches within a tree, between leaves within a branch and between leaf discs from a single leaf. The statistical form of the model is given in S1 Section B in S1 File.

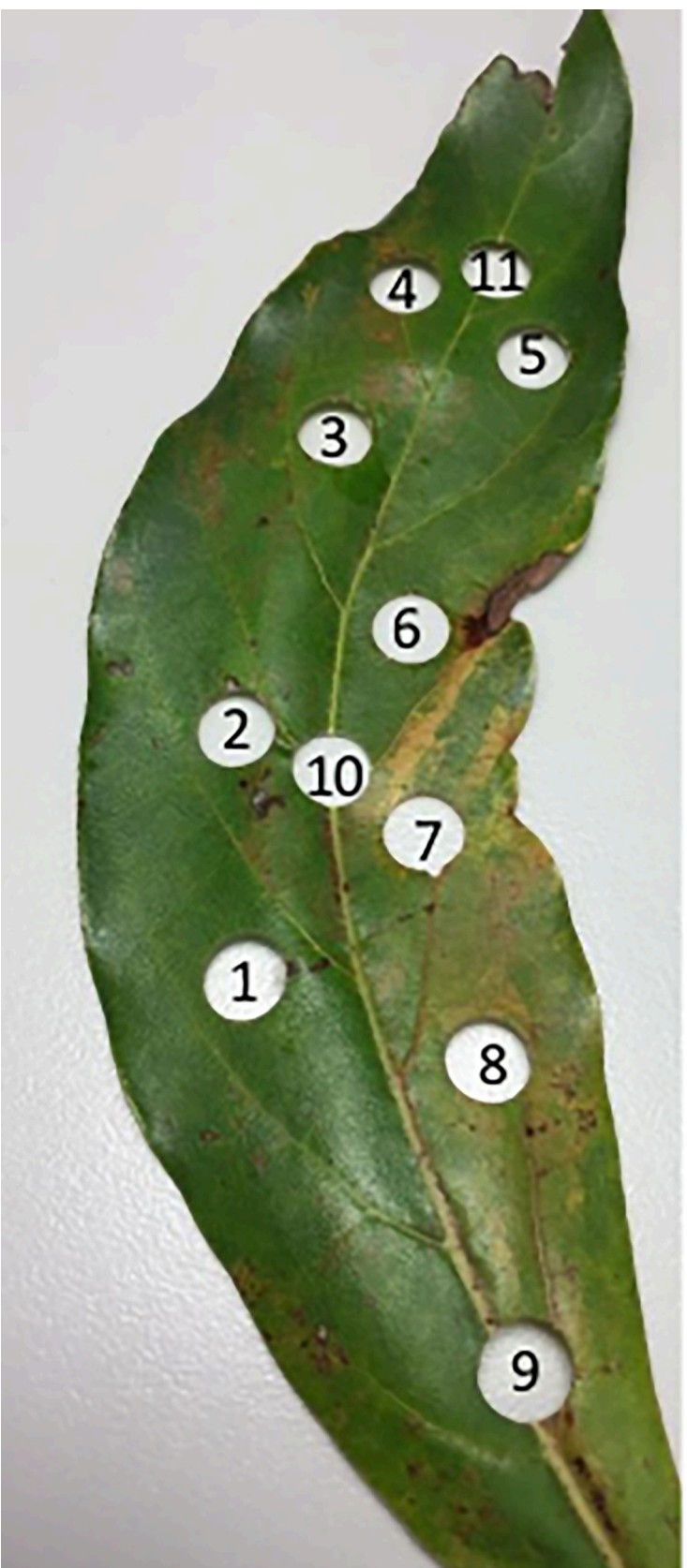

**Fig 1. Image of leaf disc locations.**

### Experiment 3: Modelling the dilution effect of batch testing on detection of ASBVd

To economize, it is necessary to pool leaves from within and amongst different trees into batches for RT-qPCR testing. Experiment 3 was designed to estimate the dilution effect of batch size by measuring the Ct value for leaf tissue composites from batches in which a single infected leaf was mixed with an increasing number of healthy leaves. A 400 mg quantity of tissue was punched from an infected leaf with predetermined Ct value of 10.55. The Ct value of the infected leaf was determined using the filter disc extraction method [17] and mixed with 0, 9, 19, and 39 to 199 by increments of punched material from healthy leaves. A second infected leaf with Ct value in the same order, 11.34, as the original leaf was used for leaf dilutions from increments of 109 healthy leaves to 199.

The relationship between the average of the Ct measurements of a batch of $n$ leaves, the average of the Ct measurements of the single infected leaf in the batch and the number of leaves in the batch is given by Eq 1 (see S1 Section C in S1 File for details):

$$\bar{C}t_{batch} - \bar{C}t_i = A \times log_{10}(n) + \varepsilon, \tag{Eq 1}$$

where $\bar{C}t_{batch}$ and $\bar{C}t_i$ denote the average Ct measurements of the batch and an infected leaf $i$ respectively; $A$ corresponds to the increase of Ct for a dilution factor of 10, and $\varepsilon$ is a measure of residual error, of standard deviation, $\sigma_\varepsilon$, which includes the intrinsic variability of the biological material and the technical variability of the assay (repeated measurements from the same biological material being denoted as technical replicates) [20]. More specifically, the variance of the average of a Ct measurement over a number, $j$, of technical replicates is the sum of terms that account for biological variations and technical measurements, with the latter proportional to the inverse of the number of technical replicates: $\sigma_\varepsilon^2 = \sigma_{bio.rep.}^2 + \sigma_{tech.rep.}^2/j$ (for details, see S1 Section C in S1 File).

### Ethics statement

In line with the PLOS ONE Observational & field-work guidelines, the authors verified at the onset that permits and approvals were not required from plant health authorities as avocado sunblotch viroid is a non-regulated pathogen and there are no restrictions or requirements related to working with ASBVd in the selected field-work locations. Verbal and email consent was obtained from the landowners and farm managers to access the privately owned avocado orchards, retain the infected trees and collect the leaf samples for the purpose of these investigations. No data were collected on individual growers or farming practices.

## Results

The aims of Experiments 1 and 2 were to study variation in viroid titer within a tree and to determine if a particular part of the tree were preferable for sampling to maximize the probability of detection of the viroid.

### Experiment 1: Leaf, branch (or octant) and tree effects on viroid titer

Experiment 1 revealed significant variation in viroid titer between the samples but there was no evidence for preferential accumulation of the viroid at a particular position(s) in the tree (see S1 Sections B.1 and B.2 in S1 File for detailed results and associated statistical analyses). For example, the height or orientation of the tree octant could not be used to predict viroid titer. Recommendations on sample selection can be drawn from these results. If four or more leaves are sampled from a tree, then ideally the leaves should be collected from different

branches or octants. However, if fewer leaves are collected, there is no justification for sampling from the top of the tree if it is costly or poses an unacceptable health risk to the worker.

## Experiment 2: Variation in viroid titer across the leaf

No preferential areas of viroid accumulation across the leaf were observed (see S1 Sections B.1 and B.3 in S1 File for detailed results and associated statistical analyses). The distribution of the viroid in a symptomatic leaf (variegated and distorted) is shown in Figs *1* & *2* and the range of Ct values across the 11 sampling locations was 14.8–18.8. The titer of viroid in the strongly variegated portion of the leaf (discs 7 and 8, Ct = 15.9 and 16.1, respectively) was only marginally higher than the adjacent greener portion of the leaf (discs 1 and 2, Ct = 17.8 and 17.3, respectively).

## Experiment 3: Modelling the dilution effect of batch testing

The simple linear model (Eq 1) was successfully fitted to a metric for the average CT difference between batch and infected leaf, and the logarithm of the sample size (Fig 3). This, in turn, allowed computation of the slope parameter, *A*, and a simple measure of the two components of the variability of the assay (i.e., taking account of variability due to intrinsic biological

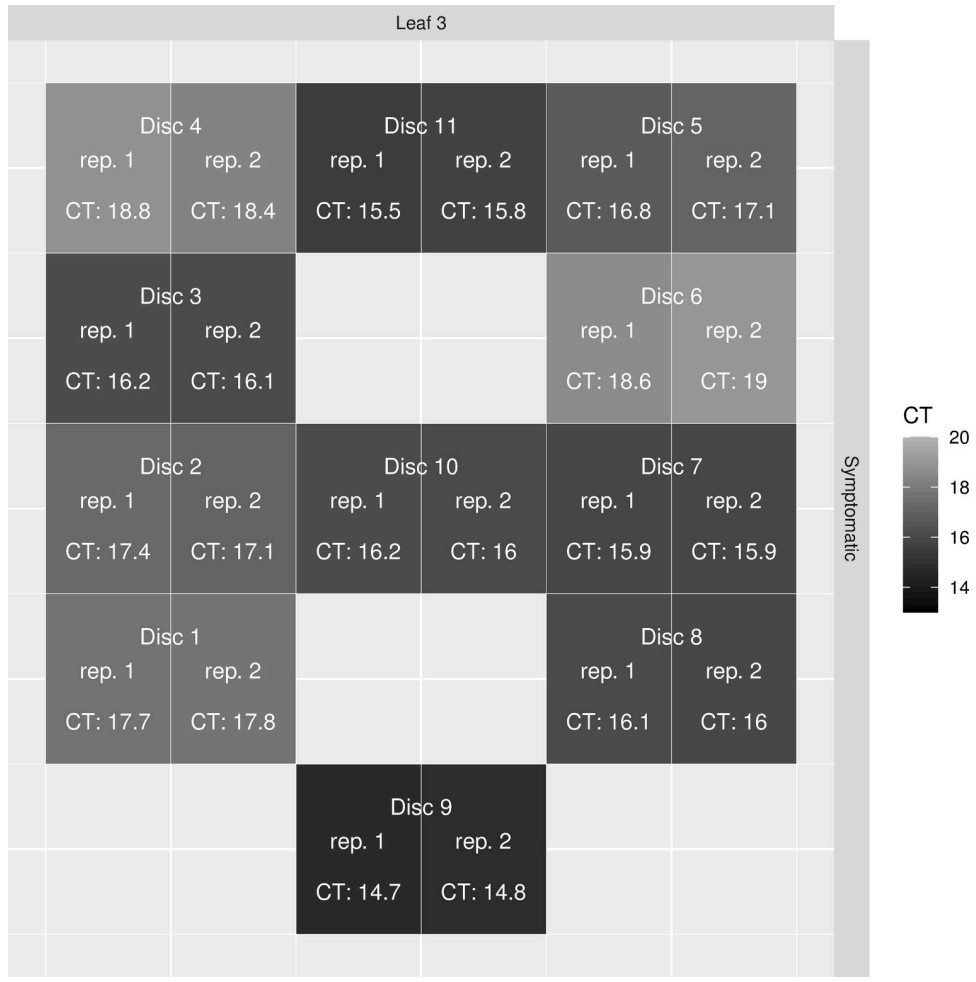

**Fig 2. Heatmap representation of Ct values from duplicate RT-qPCR assays of each disc for a specific leaf.**

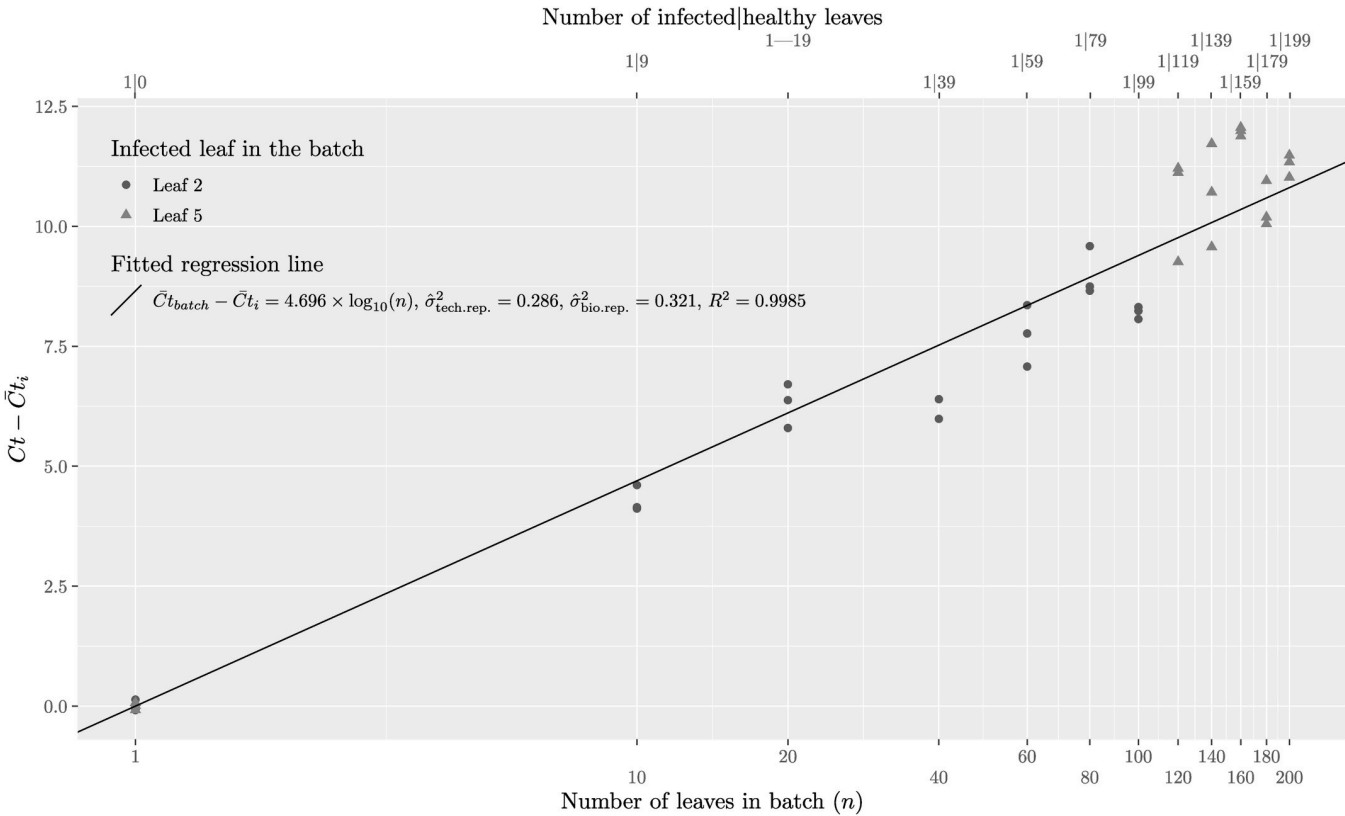

**Fig 3. Relationship between number of leaves in a mixed batch of leaves and increase in Ct where batches comprise a single ASBVd-infected leaf pooled with healthy avocado leaves.**

variation and variability associated with technical replication) for use in designing an optimal sampling strategy.

The estimates (with three significant figures) for parameters $A$, $\sigma^2_{tech.\ rep.}$ and $\sigma^2_{bio.\ rep.}$ are $\hat{A} = 4.70$, $\hat{\sigma}^2_{bio.\ rep} = 0.321$ and $\hat{\sigma}^2_{tech.\ rep} = 0.286$. Noting, the number of technical replicates was $j = 3$, we then estimate $\hat{\sigma}_\varepsilon = \left(0.321 + \frac{0.286}{3}\right)^{1/2} = 0.645$.

## Design of an optimal survey protocol

An interactive app is available at [21] that allows computation of optimal survey designs to establish pest free status for ASBVd. Each of the key parameters introduced below can be varied to reflect different sets of criteria required by regulatory authorities and logistical constraints associated with surveillance costs. The app is flexible and can be readily adapted for other systems in which there is batch sampling of leaf tissue from different plants.

Computation of an optimal sampling design to establish pest free status for ASBVd in an orchard depends upon regulatory constraints, the specifications of the diagnostic assay and orchard characteristics (Table 1). Regulatory constraints reflect international biosecurity standards in trade agreements, which are expressed in terms of thresholds of *unit level detection limit* $(1-\beta)$, *field prevalence* $(r)$ and *level of confidence* $(\alpha)$, as described by Sequeira and Griffin [22]. Note that $\beta$, $r$ and $\alpha$ are set by the regulatory authority as is the detection assay. Here we interpret these parameters for the particular case of ASBVd in which leaves are tested in batches from multiple trees. For ASBVd infecting an avocado tree, the unit level limit of

**Table 1. Index of notation for parameter values, with default values, and variables used in computing an optimal sampling design for disease-free status.**

| Category | Parameter/ variable | Default value | Definition |
|---|---|---|---|
| High level regulatory constraints | $\alpha$ | 0.95 | Confidence level (for false negative) |
| | $r$ | 0.005 | Prevalence (infected trees in target orchard) |
| | $\beta$ | 0.67 | Target percentage of healthy leaves (e.g., with Ct above $Ct_{target}$) for infected trees |
| Additional regulatory constraint | $Ct_{target}$ | 15 | Ct threshold used to define $\beta$ |
| | $\alpha_b$ | 0.99 | Batch level accepted risk |
| PCR characteristics | $j$ | 1 | Number of technical replicates |
| | $A$ | 4.7 | Effect of a dilution by 10 on the Ct value |
| | $Ct_{detect}$ | 25 | Threshold used for detection |
| | $\hat{\sigma}_\varepsilon^2$ | 0.8 | Variance of the Ct measurement at the batch level. This variance accounts for variation related to differences between biological samples and technical measurements $\hat{\sigma}_\varepsilon^2 = \hat{\sigma}_{bio.rep.}^2 + \hat{\sigma}_{tech.rep.}^2/j$ |
| Orchard characteristics | $N$ | 6000 | Population size |
| Sampling characteristics (to be determined by optimisation) | $n_0$ | - | Number of batches |
| | $n_1$ | - | Number of donor trees in one batch |
| | $n_2$ | - | Number of leaves per donor tree |
| Cost function | $Cost(n_0, n_1, n_2)$ | $n_0$ | Cost associated with a specific allocation. By default, the costs are equated with the number of batches |

detection of infected leaves per tree $(1-\beta)$ is the percentage of leaves on a tree with a defined amount or copy number of the viroid. We define a target threshold for the Ct value ($Ct_{target}$) that is lower than the *detection Ct* (denoted $Ct_{detect}$), which corresponds to the minimal Ct value that can be obtained for leaves that do not contain the viroid. The prevalence corresponds to the percentage of trees in an orchard with a unit level of infection above the unit level detection limit. The confidence level is the probability of detecting the viroid in a field with a given prevalence. Default values for the key parameters are summarized in Table 1. The default value for β = 0.67 is similar to that used by the Ministry of Agriculture in New Zealand [10]. The default values for the PCR parameters that we use reflect the estimates that were obtained from Experiment 3. They are used here for illustrative purposes and can be adjusted by regulatory authorities. The accompanying App allows flexibility in setting all the parameters [21].

## Calculating optimal strategies

We consider a population of trees of size $N$, such that at least $N \times r$ trees each have a proportion higher than $1-\beta$ of leaves with a Ct value smaller than $Ct_{detect}$. When (i) $n_0$ batches are being tested, (ii) each batch contains exactly $n_2$ leaves from each of $n_1$ different trees, (iii) leaves from the same tree are used for at most one batch, (iv) the residuals can be treated as independent Gaussian variables with zero mean and equal variances, (v) the sampling design is simple random sampling or systematic sampling, then the upper boundary for the probability of not detecting ASBVd (i.e., the Type II risk) is approximately given by:

$$\text{Maj}Risk(N, r, n_0, n_1, n_2, \beta)$$
$$= \left( \sum_{t=0}^{n_1} \binom{n_1}{t} r^t (1-r)^{n_1-t} \sum_{l=0}^{t \times n_2} \binom{n_2 \times t}{l} \beta^{t \times n_2 - L} (1-\beta)^L \left( 1 - \Phi \left( \frac{Ct_{detect} - Ct_{target} - A \times (log_{10}(n_1 \times n_2) - log_{10}(1))}{\sigma_\varepsilon} \right) \right) \right) \right)^{n_0},$$

where $\Phi$ is the cumulative normal distribution function (see S1 Section D in S1 File for details).

## Cost function

The cost function is the total cost of the operation and is a function of $n_0$, $n_1$ and $n_2$ denoted by $Cost(n_0, n_1, n_2)$. The cost function depends upon resourcing and it is reasonable to assume the cost function is an additive weighted function of $n_0$, $n_1$, $n_2$.

## Optimal allocation

Optimal sampling parameters are defined as the ones that minimize $Cost(n_0, n_1, n_2)$ under the constraints that the approximated risk function (computed with estimates $\hat{A}$ and $\hat{\sigma}_\varepsilon$ for A and $\sigma_\varepsilon$) is less than $\alpha$, and that the number of leaves per batch is such that the probability of detecting a single infected leaf in a batch is above a batch level confidence parameter, $\alpha_b$. Details of the computation of the optimal sample allocation are given in Supporting Information S1 Section E in S1 File.

 We investigated optimal designs including pooling of leaf samples amongst trees in a multi-stage framework (Table 2). The sampling mechanism is characterized by three parameters: the total number of batches (denoted $n_0$), the number of trees per batch (denoted $n_1$) and the number of leaves sampled from each tree (denoted $n_2$), with a simple cost function in which cost is dominated by the number of batches (i.e. $Cost(n_0, n_1, n_2) = n_0$). The optimal allocation under an assumption of the cost being a function only of the number of batches ($Cost(n_0, n_1, n_2) = n_0$) was computed for a set of baseline parameters (row 1 in Table 2), then re-computed after changing each parameter (Table 2).

 When the cost is only a function of the number of batches, the optimization results indicate preference for saturating the number of leaves per batch ($n_1 \times n_2$). The maximal number of leaves per batch is a function of the RT-qPCR parameters, as well as $\alpha_b$ and $Ct_{target}$ and is given in in Supporting Information S1 Eqn 4 in S1 File. Then the optimization results indicate that the trade-off between $n_1$ and $n_2$ favours $n_1$ over $n_2$: i.e., sampling a large number of trees with

**Table 2. Optimal sample sizes for a selection of inputs.**

| Inputs | | | | No. trees in orchard | PCR testing: Detection threshold ($ct_{detect}$) standard deviation of Ct measurements ($\hat{\sigma}_\varepsilon$), dilution assay ($\hat{A}$) effect, and batch level confidence ($\alpha_b$) | | | | Optimal sample designs | | | | |
|---|---|---|---|---|---|---|---|---|---|---|---|---|---|
| Regulations: Confidence level ($\alpha$), percentage of healthy leaves in infected trees ($\beta$), prevalence ($r$), target Ct | | | | | | | | | Optimal number of batches ($n_0^*$), trees per batch ($n_1^*$) and leaves per tree ($n_2^*$) | | | Total number of leaves per batch ($n_1^* \times n_2^*$) at the optimum and limit L of leaves per batch. | |
| $\alpha$ | $\beta$ | $r$ | $Ct_{target}$ | $N$ | $Ct_{detect}$ | $\hat{\sigma}_\varepsilon$ | $\hat{A}$ | $\alpha_b$ | $n_0^*$ | $n_1^*$ | $n_2^*$ | $n_1^* \times n_2^*$ | $L$ |
| 0.95 | 0.67 | 0.005 | 15 | 6000 | 25 | 0.8 | 4.7 | 0.99 | 35 | 53 | 1 | 53 | 53 |
| 0.99* | 0.67 | 0.005 | 15 | 6000 | 25 | 0.8 | 4.7 | 0.99 | 54 | 53 | 1 | 53 | 53 |
| 0.95 | 0.90* | 0.005 | 15 | 6000 | 25 | 0.8 | 4.7 | 0.99 | 131 | 17 | 3 | 51 | 53 |
| 0.95 | 0.67 | 0.001* | 15 | 6000 | 25 | 0.8 | 4.7 | 0.99 | 219 | 25 | 2 | 50 | 53 |
| 0.95 | 0.67 | 0.001 | 20* | 6000 | 25 | 0.8 | 4.7 | 0.99 | 456 | 4 | 1 | 4 | 4 |
| 0.95 | 0.67 | 0.001 | 15 | 12000* | 25 | 0.8 | 4.7 | 0.99 | 35 | 53 | 1 | 53 | 53 |
| 0.95 | 0.67 | 0.005 | 15 | 6000 | 19* | 0.8 | 4.7 | 0.99 | 908 | 2 | 1 | 2 | 2 |
| 0.95 | 0.67 | 0.005 | 15 | 6000 | 25 | 1.6* | 4.7 | 0.99 | 88 | 21 | 1 | 21 | 21 |
| 0.95 | 0.67 | 0.005 | 15 | 6000 | 25 | 0.8 | 6.2* | 0.99 | 92 | 20 | 1 | 20 | 20 |
| 0.95 | 0.67 | 0.005 | 15 | 6000 | 25 | 0.8 | 4.7 | 0.999* | 47 | 39 | 1 | 39 | 39 |
| 0.95 | 0.67 | 0.005 | 15 | 6000 | 25 | 0.8 | 4.7 | 0.95* | 28 | 68 | 1 | 68 | 70 |

*Indicates parameter has been changed compared with base line (first row).

low numbers of leaves per tree is preferred, and often only a single leaf from each tree is required. Table 2 shows that the optimal number of leaves per tree is ordinarily 1, except in the case where the targeted percentage of infected leaves in the tree is comparatively small (e.g., $\beta$ = 0.9, corresponding to 10% leaves infected in an infected tree, in Table 2). Sampling one or a few leaves per tree carries a risk of failing to detect an infected tree, albeit with low probability $1-\alpha_b$ = 0.01 (Table 2) and our analysis indicates that it is better to sample more trees for the same cost than to sample more leaves and fewer trees in testing for freedom from infection in an orchard. This holds when the proportion of infected leaves on a tree is high, but optimal designs indicate sampling more leaves per tree when the proportion decreases (cf. (1- $\beta$) = 0.1 in row 3 in Table 2). Reducing the critical value for prevalence in the target population, switches the balance towards sampling more batches with fewer trees per batch Table 2). Some parameters, when changed, impact the maximal number of leaves per batch, which may need to be compensated for to reach the required level of confidence by increasing the number of batches (cf. changing $Ct_{\text{detect}}$, $Ct_{\text{target}}$ and $\hat{\sigma}_\varepsilon$ in Table 2). Other parameters do not impact the maximal number of leaves per tree but impact directly the required level of confidence or the prevalence (see S1 Eqn 4 in S1 File for further details).

We chose a default value of $N$ = 6,000 trees for a target orchard (Tables 1 and 2). Increasing $N$ to 12,000 trees does not affect the optimal sampling protocol in terms of $n_0$, $n_1$, and $n_2$, (cf. row 6 with row 1 in Table 2). Although this result at first appears counter intuitive, it reflects an asymptotic influence of orchard size in which drawing from a population of 6,000 trees with a fixed rate ($r$) of infected trees is similar to drawing from an infinite population of trees with the same rate of infected trees in the sense that the probability distribution of the number of infected trees in the sample is approximately the same. Setting criteria in relation to the field size $N$ and the prevalence $r$ is under the control of a regulatory authority. The regulatory authority may choose to assess the presence in an extensive large area that includes a large number of trees or to constrain areas for detection to a maximum size, say 6000 trees for separate assessment.

The interactive app available at [21] allows all the key parameters including cost functions to be changed (see also selective screen shots S1 Figs 3 & 4 in S1 File).

Optimal sample sizes for a selection of different cost functions show marked changes in the balance of sampling effort depending upon the weighting given in cost functions to batches, trees and leaves (Table 3). The baseline (Table 3, row 1 of results) is the same as for Table 2, row 1 of results. When the cost of sampling trees dominates (Table 3, row 2 of results), and the

**Table 3. Optimal sample sizes for different cost functions.**

| Specific Inputs | | Outputs Optimal number of batches ($n_0^*$), trees per batch ($n_1^*$) and leaves per tree ($n_2^*$) | | | Practical outputs At the optimum, total number of selected trees ($n_0^* \times n_1^*$), total number of selected leaves ($n_0^* \times n_1^* \times n_2^*$) and cost. | |
|---|---|---|---|---|---|---|
| Cost function | $Cost(n_0, n_1, n_2) = \ldots$ | $n_0^*$ | $n_1^*$ | $n_2^*$ | $n_0^* \times n_1^*$ | $n_0^* \times n_1^* \times n_2^*$ |
| Number of batches | $n_0$ | 35 | 53 | 1 | 1855 | 1855 |
| Number of trees | $n_0 \times n_1$ | 598 | 1 | 19 | 598 | 11362 |
| Number of leaves | $n_0 \times n_1 \times n_2$ | 1815 | 1 | 1 | 1815 | 1815 |
| Combination | $n_0 + (n_0 \times n_1)$ | 104 | 6 | 8 | 660 | 3960 |
| Combination | $n_0 + 2 \times (n_0 \times n_1 \times n_2)$ | 55 | 33 | 1 | 1815 | 1815 |
| Custom | $n_0 + (Inf \times (n_1 \times n_2) > 10)))$ | 182 | 10 | 1 | 182 | 1820 |

Optimal allocation is given for default parameters as specified in Table 1

costs related to PCR and collecting leaves are negligible, the optimal strategy reflects a trade-off between $n_1$ and $n_2$ with a large number (19) of leaves collected per tree but only one tree per batch. The outcome is a very large number ($n_1$ x $n_2$) of leaves to collect (Table 3).

If the cost were dominated by the number of leaves collected (Table 3, row 3 of results), there is no gain in pooling leaves together, then each leaf is tested individually and the number of PCR tests is impractically large. When the cost is a combination of the PCR, trees and leaves sampling costs, (Table 3, rows 4 and 5 of results), the optimal allocation reflects a trade-off between the different costs.

The accompanying app allows custom cost functions to be defined, (*cf.* Table 3, row 6 of results), which may reflect extra constraints, as, for example, putting an infinite cost on allocations with more than 10 trees per batch, which is equivalent to putting a hard constraint on the number of trees per batch.

We note that the total number of leaves sampled often approaches a minimum of 1,815, unless the costs involve the total number of trees but not the number of leaves. This is consistent with the results from Table 2 that minimizing the number of batches ($n_o$) favours the number of trees per batch rather than the number of leaves per tree.

## Discussion

We have developed a scalable protocol for use in optimizing sampling strategies to establish pest free status from ASBVd in avocado orchards. The protocol, which is supported by an interactive app [21], integrates statistical considerations of multistage sampling of trees in orchards with a RT-qPCR assay allowing for detection of infection in pooled samples of leaves taken from multiple trees. While the approach was designed with ASBVd in mind, we note that it has broad applicability for a wider range of plant pathogens in which hierarchical sampling of a target population is coupled with pooling of material prior to assay. For example, in sampling for regional pest-free status of an agricultural or horticultural crop, samples may be collected within fields and aggregated amongst fields to minimize the costs of running the diagnostic assay.

Our analyses for ASBVd take account of high-level regulatory constraints as well as practical constraints associated with the assay used to detect a positive response (Table 1). Regulatory constraints would normally be set by a government or other regulatory authority. The constraints include an arbitrary low-limit of prevalence ($r$) against which an orchard is declared pest free with a confidence level ($\alpha$) for a false negative assertion. It is also necessary to set a regulatory constraint for the target percentage of infected leaves in an infected tree (1 - $\beta$). The default value for $\beta$ = 0.67 used in our analyses was chosen to match the value set by the New Zealand Ministry of Agriculture and Forestry [10], which is the only biosecurity regulatory authority to have addressed the question as to what constitutes acceptable evidence of pest freedom with regards to ASBVd [23]. Critical parameters related to the RT-qPCR assay include not only the usual Ct threshold ($Ct_{threshold}$) used for detection but an additional Ct target threshold ($Ct_{target}$) related to $\beta$.

We used a simple experiment to demonstrate a linear relationship for the dilution effect of mixing healthy with an infected leaf on Ct values. The slope and variance from the relationship were used in the calculation of optimal designs. The experiments were designed as proof of concept using small numbers of infected plants. We recommend recalibration of experiment 3 using larger numbers of leaves to improve the estimates for $\hat{A}$ and $\hat{\sigma}$.

Analysis of the variability in Ct measurements indicated that the distribution of ASBVd varied between branches and octants within infected trees but not in a consistent way to identify viroid 'hotspots' that could be preferentially sampled in order to maximize the chances of

detecting the viroid in an infected tree. Similar results applied to detection of the viroid within infected leaves. The practical consequence is that to detect the presence of the pathogen in a field, for a given number of collected leaves, the number of donor trees should be maximized in preference to sampling more than one leaf per tree (Table 2). From a practical perspective, it is not feasible to sample mature avocado trees in commercial orchards containing thousands of trees from any layer other than at ground height. However, for small populations of high-value avocado trees, such as those in germplasm collections or in multiplication blocks used for propagation, multiple leaves from different parts of the tree should be sampled, as is normal protocol in many countries such as Australia and South Africa.

The approach we followed here in defining two thresholds $((1-\beta)$ and $Ct_{target})$ is analogous to the general situation described in official European Food Safety Authority guidelines [14]. The guidelines [14] list four parameters that govern the sampling phase from which sample size can be computed: the confidence level, the field level prevalence, the field size, and a term referred to as 'method sensitivity' whereby the risk manager needs to assess the sensitivity of the assay method in detecting the target pathogen. Our parameters for the qRT-PCR $A$ and $\sigma_\varepsilon$ are used to characterize the method sensitivity and need to be estimated beforehand. In addition to setting a criterion for field prevalence, non-uniform distribution of ASBVd within a tree requires extra criteria to be set for prevalence within plants, which are defined by the thresholds $(1-\beta)$ and $Ct_{target}$.

In the absence of auxiliary information on the trees (such as root stock, variety, symptoms of infection or presence of infected trees in a vicinity), we recommend a simple protocol involving systematic sampling of trees to ensure even coverage of the target orchard followed by simple random sampling [24] of units at each subsequent stage. The simulation tool we provide allows comparison of the risk of false negatives at the orchard level for simple random or systematic sampling. Use of the app shows that systematic sampling is optimal.

Prior to doing any diagnostic assays for ASBVd, trees in the orchard should be inspected for symptoms and when recognized, the diseased trees should be individually tested along with neighboring trees either in the same or adjacent row, since there is evidence for transmission of the viroid by natural root grafting [25]. The appearance of symptoms such as leaf bleaching or variegation may indicate that the infection is only at an early phase, in which case the symptomatic leaves should be sampled as they are more likely to contain a higher viroid titer [25]. However, according to the longitudinal study of Semancik and Szychowski [25], this initial acute phase of infection is followed by a chronic phase, when foliar symptoms disappear and the viroid becomes uniformly distributed around the tree at a high titer. The trees in our study were typical of the chronic phase of infection, at which point it is likely that leaves could be collected from any point on the tree with an equal chance of detecting the viroid. The optimal design protocol should be applied to orchards without any signs of sunblotch disease. Ultimately, the surest way to guarantee that an orchard is free of ASBVd is to use certified planting material that has been propagated using seed and budwood from mother trees that have been tested and shown to be free of ASBVd. With basic orchard hygiene, such as using dedicated pruning equipment for that particular orchard, it would be extremely unlikely that the viroid would be freshly introduced into the orchard and there should not be a need to do anymore testing for duration of the trees' life.

## Source code and data

All data and code were made available in the open access Gitlab repository ASBVd_Detection [26] as well as in the S2 File in the form of an R package. The code execution was demonstrated in the dedicated Gitlab pages [27].

## Supporting information

**S1 File. Additional figures, mathematical formulas and presentation of the interactive computation tool.**
(PDF)

**S2 File. ASBVdDetection.** The archive file asbvd-detect-r.tar.gz is an R package that contains the source code and data used for all the statistical analysis, as well as the source code for the interactive tool [22]. It is a copy of the source code available at [26].
(GZ)

## Acknowledgments

We thank the anonymous avocado grower in South-east Queensland for practical support and engagement. We also thank the anonymous avocado grower in Mpumalanga, South Africa, for allowing access to orchards under their management and Lebogang Motaung and Ayanda Msweli for technical assistance in the laboratory. We thank Dr Alison Scott-Brown for assistance with literature retrieval and support with the final editing process and acknowledge the helpful comments given by the Editor and the two anonymous reviewers on an earlier version of this manuscript.

## Author Contributions

**Conceptualization:** D. B. Bonnéry, L. -S. Pretorius, A. E. C. Jooste, A. D. W. Geering, C. A. Gilligan.

**Data curation:** D. B. Bonnéry, L. -S. Pretorius, A. E. C. Jooste.

**Formal analysis:** D. B. Bonnéry, L. -S. Pretorius, A. E. C. Jooste, A. D. W. Geering, C. A. Gilligan.

**Funding acquisition:** A. D. W. Geering, C. A. Gilligan.

**Investigation:** D. B. Bonnéry, L. -S. Pretorius, A. E. C. Jooste, A. D. W. Geering, C. A. Gilligan.

**Methodology:** D. B. Bonnéry, L. -S. Pretorius, A. E. C. Jooste, A. D. W. Geering, C. A. Gilligan.

**Project administration:** A. D. W. Geering.

**Resources:** A. D. W. Geering.

**Software:** D. B. Bonnéry.

**Supervision:** A. D. W. Geering, C. A. Gilligan.

**Validation:** A. D. W. Geering, C. A. Gilligan.

**Visualization:** D. B. Bonnéry, L. -S. Pretorius, A. E. C. Jooste.

**Writing – original draft:** D. B. Bonnéry, L. -S. Pretorius, A. E. C. Jooste, A. D. W. Geering, C. A. Gilligan.

**Writing – review & editing:** D. B. Bonnéry, L. -S. Pretorius, A. E. C. Jooste, A. D. W. Geering, C. A. Gilligan.

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
