## [Decision Letter · Decision Letter 0]

11 Jan 2023

PONE-D-22-30012Rational design of a sampling protocol for detection of a subcellular plant pathogen to demonstrate area freedom in commercial orchardsPLOS ONE

Dear Dr. Bonnéry,

Thank you for submitting your manuscript to PLOS ONE. After careful consideration, we feel that it has merit but does not fully meet PLOS ONE’s publication criteria as it currently stands. Therefore, we invite you to submit a revised version of the manuscript that addresses the points raised during the review process.

This is a research protocol paper. I expect widely acceptable details on method(s). It seems to me you had it but needs to clarify some details as one of the reviewers showed. I think your title is beyond what you suggest in general although the method might be used for other crops but needs to be verified as a protocol. I hope you will be able to make clearer statements.

Please submit your revised manuscript by Feb 25 2023 11:59PM. If you will need more time than this to complete your revisions, please reply to this message or contact the journal office at plosone@plos.org. Please include the following items when submitting your revised manuscript:A rebuttal letter that responds to each point raised by the academic editor and reviewer(s). You should upload this letter as a separate file labeled 'Response to Reviewers'.A marked-up copy of your manuscript that highlights changes made to the original version. You should upload this as a separate file labeled 'Revised Manuscript with Track Changes'.An unmarked version of your revised paper without tracked changes. You should upload this as a separate file labeled 'Manuscript'.

We look forward to receiving your revised manuscript.

Kind regards,

Ahmet Uludag, Ph.D.

Academic Editor

PLOS ONE

Journal Requirements:

“This project has been funded by Hort Innovation, using the avocado research and

development levy and contributions from the Australian Government under Project

AV18007 – Avocado sunblotch viroid survey. Hort Innovation is the grower owned, not-forprofit research and development corporation for Australian horticulture.”

“This project has been funded by Hort Innovation (https://www.horticulture.com.au/), using the avocado research and development levy and contributions from the Australian Government under Project AV18007 – Avocado sunblotch viroid survey. Hort Innovation is the grower owned, not-for-profit research and development corporation for Australian horticulture.  The  grant was received by A. G.  The funders had no role in study design, data collection and analysis, decision to publish, or preparation of the manuscript.”

Additional Editor Comments:

Two reviewers have suggested different decisions deeply. I would like to have your responds on reviewers' comments, especially from one who has rejected. I definitely agree with editor on too broad title you have. I think you need to explain methodic problems carefully.

Reviewers' comments:

Reviewer's Responses to Questions

**Comments to the Author**

1. Is the manuscript technically sound, and do the data support the conclusions?

Reviewer #1: Yes

Reviewer #2: Partly

2. Has the statistical analysis been performed appropriately and rigorously? 

Reviewer #1: Yes

Reviewer #2: I Don't Know

3. Have the authors made all data underlying the findings in their manuscript fully available?

Reviewer #1: Yes

Reviewer #2: No

4. Is the manuscript presented in an intelligible fashion and written in standard English?

Reviewer #1: Yes

Reviewer #2: No

5. Review Comments to the Author

Reviewer #1: Dear Editor,

The study presents the results of original research and its results reported have not been published elsewhere. Experiments, statistics, and other analyses are performed to a high technical standard and are described in sufficient detail. Conclusions are presented in an appropriate fashion and are supported by the data. The article is presented in an intelligible fashion and is written in good English. The research meets all applicable standards for the ethics of experimentation and research integrity. The article adheres to appropriate reporting guidelines and community standards for data availability. However, I suggest some minor revisions. Introduction of the paper is too long and readers can get bored/tired. Please, shorten it if applicable. And, my decision is to accept the article after the minor revisions.

Reviewer #2: This manuscript describes a protocol for field sampling and detection of ASBVd that can be used as a regulatory procedure which, if samples from a field or region are negative for ASBVd, can support export of avocado to a country that does not accept import of avocado from a region or field that has ASBVd. I believe the title is too broad (e.g. subcellular plant pathogen) and should be specific for avocado and ASBVd. Many other viruses and viroids can be considered subcellular and the filter disk capture of the nucleic acid template may not be considered robust enough as part of a regulatory detection platform. Moreover, the molecular description of ASBVd is not adequately described to provide background needed to evaluate template preparation and RT-qPCR methods used. I do not like the term “negotiable” regarding an export protocol. Rather, it is based on solid data or a series of criteria that are “mutually” accepted. This and specific comments included in the attachment

6. PLOS authors have the option to publish the peer review history of their article (what does this mean?). If published, this will include your full peer review and any attached files.

Reviewer #1: No

Reviewer #2: No

---

## [Author Response · Author response to Decision Letter 0]

1 Mar 2023

We thank the editor and the reviewer for their review.

We have provided a detailed answer in the rebuttal letter attached to this resubmission,

---

## [Decision Letter · Decision Letter 1]

27 Mar 2023

Rational design of a survey protocol for avocado sunblotch viroid in commercial orchards to demonstrate pest freedom

PONE-D-22-30012R1

Dear Dr. Bonnery,

We’re pleased to inform you that your manuscript has been judged scientifically suitable for publication and will be formally accepted for publication once it meets all outstanding technical requirements.

Kind regards,

Ahmet Uludag, Ph.D.

Academic Editor

PLOS ONE

Additional Editor Comments (optional):

Thank you very much for your very good revision. Please consider some editorial points by a reviewer when you have received a copy before publishing.

Reviewers' comments:

Reviewer's Responses to Questions

**Comments to the Author**

1. If the authors have adequately addressed your comments raised in a previous round of review and you feel that this manuscript is now acceptable for publication, you may indicate that here to bypass the “Comments to the Author” section, enter your conflict of interest statement in the “Confidential to Editor” section, and submit your "Accept" recommendation.

Reviewer #1: All comments have been addressed

Reviewer #2: All comments have been addressed

2. Is the manuscript technically sound, and do the data support the conclusions?

Reviewer #1: Yes

Reviewer #2: Yes

3. Has the statistical analysis been performed appropriately and rigorously? 

Reviewer #1: Yes

Reviewer #2: Yes

4. Have the authors made all data underlying the findings in their manuscript fully available?

Reviewer #1: Yes

Reviewer #2: Yes

5. Is the manuscript presented in an intelligible fashion and written in standard English?

Reviewer #1: Yes

Reviewer #2: Yes

6. Review Comments to the Author

Reviewer #1: Unlike the other referee, I have previously stated that the make can be accepted after minor revision. However, the constructive suggestions of both the other referee and the editorial office on the article cannot be ignored. It is seen that the authors responded to all the points mentioned and revised the article as desired. I think the article can be published in this state.

Reviewer #2: This manuscript has been adequately revised from the original submission. It describes a protocol for field sampling and detection of ASBVd that can be used as a regulatory procedure which, if samples from a field or region are negative for ASBVd, can support export of avocado to a country that does not accept import of avocado from a region or field that has ASBVd. This reviewer does understand novelty of this work is in the design, development, and implementation of a statistical protocol to optimize sampling for pest freedom and not the standard qRT-PCR protocols for its detection. My comments are either to bring out methods that were used but not mentioned or to improve the final procedures that will be presented to the regulatory bodies for adaptation. This paper is acceptable for publication.

7. PLOS authors have the option to publish the peer review history of their article (what does this mean?). If published, this will include your full peer review and any attached files.

Reviewer #1: No

Reviewer #2: No

---

## [Editor Report · Acceptance letter]

30 Mar 2023

PONE-D-22-30012R1 

Rational design of a survey protocol for avocado sunblotch viroid in commercial orchards to demonstrate pest freedom 

Dear Dr. Bonnéry:

I'm pleased to inform you that your manuscript has been deemed suitable for publication in PLOS ONE. Congratulations! Your manuscript is now with our production department. 

Kind regards, 

on behalf of

Dr. Ahmet Uludag 

Academic Editor

PLOS ONE